# A Sweet Potato MYB Transcription Factor *IbMYB330* Enhances Tolerance to Drought and Salt Stress in Transgenic Tobacco

**DOI:** 10.3390/genes15060693

**Published:** 2024-05-26

**Authors:** Chong Wang, Jian Lei, Xiaojie Jin, Shasha Chai, Chunhai Jiao, Xinsun Yang, Lianjun Wang

**Affiliations:** 1Institute of Food Crops, Hubei Academy of Agricultural Sciences, Hubei Key Laboratory of Food Crop Germplasm and Genetic Improvement, Key Laboratory of Crop Molecular Breeding, Ministry of Agriculture and Rural Affairs, Wuhan 430064, China; wangchong19409@163.com (C.W.); leijian2006@hbaas.com (J.L.); xiaojiejin@hbaas.com (X.J.); chaishasha2008@hbaas.com (S.C.); jiaoch@hotmail.com (C.J.); 2Crop Institute of Jiangxi Academy Agricultural Sciences, Nanchang 330200, China; 3College of Agriculture, Yangtze University, Jingzhou 434025, China

**Keywords:** *IbMYB330*, sweet potato, drought and salt tolerance, gene overexpression, tobacco

## Abstract

MYB transcription factors (TFs) play vital roles in plant growth, development, and response to adversity. Although the MYB gene family has been studied in many plant species, there is still little known about the function of R2R3 MYB TFs in sweet potato in response to abiotic stresses. In this study, an R2R3 MYB gene, *IbMYB330* was isolated from sweet potato (*Ipomoea batatas*). *IbMYB330* was ectopically expressed in tobacco and the functional characterization was performed by overexpression in transgenic plants. The IbMYB330 protein has a 268 amino acid sequence and contains two highly conserved MYB domains. The molecular weight and isoelectric point of IbMYB330 are 29.24 kD and 9.12, respectively. The expression of *IbMYB330* in sweet potato is tissue-specific, and levels in the root were significantly higher than that in the leaf and stem. It showed that the expression of *IbMYB330* was strongly induced by PEG-6000, NaCl, and H_2_O_2_. Ectopic expression of *IbMYB330* led to increased transcript levels of stress-related genes such as *SOD*, *POD*, *APX*, and *P5CS*. Moreover, compared to the wild-type (WT), transgenic tobacco overexpression of *IbMYB330* enhanced the tolerance to drought and salt stress treatment as CAT activity, POD activity, proline content, and protein content in transgenic tobacco had increased, while MDA content had decreased. Taken together, our study demonstrated that *IbMYB330* plays a role in enhancing the resistance of sweet potato to stresses. These findings lay the groundwork for future research on the R2R3-MYB genes of sweet potato and indicates that *IbMYB330* may be a candidate gene for improving abiotic stress tolerance in crops.

## 1. Introduction

There are many environmental factors that limit crop production and quality in the worldwide [1]. Drought and high salinity are two universal abiotic stressors that lead to drastic economic losses in agriculture [2,3]. As the world population develops by leaps and bounds and the frequent occurrence extreme weather increases, traditional farming faces huge threats; hence, it is important to develop crops with higher levels of drought and salt tolerance.

In plants, numerous mechanisms are involved in responding and adapting to abiotic stress conditions. In response to adversity, a variety of stress-related genes in plants encode stress-resistant proteins, antioxidants, and osmotic enzymes and thus play important roles in the defense against adversity [4]. Multiple types of transcription factor (TF) families, including MYB, NAC, MADS-box, bZIP, WRKY, and AP2/ERF, play important roles in regulating target proteins and genes in the metabolic networks of plants responding to stress. Among the transcription factor families in plants, the MYB (*v-myb* avian myeloblastosis viral oncogene homolog) gene is one of the largest groups, and multiple studies have now shown that MYB proteins play diverse roles in the responses to abiotic stresses, such as drought, salt, and cold stresses [5,6]. MYB proteins are characterized by the presence of one to four MYB repeats of 50–52 amino acids forming three *α*-helices. Based on the MYB repeat number and the identity of the MYB repeats, MYB proteins have been subdivided into four major classes: 1R-MYB, R2R3-MYB, R1R2R3-MYB, and 4R-MYB [6,7]. So far, large amounts of MYB TFs have been identified in plant species, such as *Arabidopsis*, *Gossypium Hirsutum*, and *Dendrobium catenatum* [6,8,9]. It has been found that many plants encode more than 100 R2R3-MYB TFs; for example, in *Arabidopsis*, 126 R2R3 MYB TFs have been identified [10,11]. Whole-genome analysis of plants shows that several plant species have R2R3 MYB TFs amount larger than *Arabidopsis*, for example, *Solanum lycopersicum* (127 members), *Medicago truncatula* (155 members), *Zea mays* (157 members), and *Populus trichocarpa* (192 members) [12,13,14,15]. R2R3 MYB TFs are crucial for regulating plant biochemical processes such as specialized metabolism, development, and responses to biotic and abiotic stresses [16,17,18]. Furthermore, multitudinous experiments confirm that R2R3 MYB TFs participate in the response to drought, salinity, and high temperature [19]. In *Arabidopsis thaliana*, overexpression *AtMYB2* enhanced the ABA-inducible gene expression under drought stress [20]. Transgenic *A. thaliana* overexpression *AtMYB4* enhanced tolerance to drought and salt stress compared to wild-type plants [21]. AtMYB32, AtMYB44, AtMYB49, AtMYB52, AtMYB61, and AtMYB96 responded to drought and salt stress in transgenic plants by regulating different biological metabolic pathways [21,22,23,24,25,26]. In rice, *OsMYB4*, *OsMYB60*, and *OsMYB55* were involved in drought and salt stress [27,28,29]. In *Nicotiana tabacum*, a R2R3-MYB transcription factor *NtMYB330* was isolated and characterized. This study reveals that NtMYB330 specifically regulates PA biosynthesis via formation of the MBW complex in tobacco flowers and affects germination through adjustment of PA concentrations and ABA/GA signaling in tobacco seeds [30]. The study found that R2R3-MYB MYB330 may be involved in the control of wilt, fruit ripening, seed development, and root development in *Psidium guajava* × *Psidium molle* [31]. Despite many R2R3 MYB TFs having been isolated and identified in various model plants, much is still unknown about the function of R2R3-MYB TFs.

Sweet potato (*Ipomoea batatas* (L.) Lam.) is the seventh most important food crop in the world [32]. The annual global yield of sweet potato is approximately 113 million tons, and it is an important feed, industrial raw material, and energy crop [33,34]. Sweet potato is significant for food security; besides, it has become an affordable source of dietary calories, protein, minerals, vitamins, and polyphenols, particularly in underdeveloped areas [35]. The productivity of sweet potato is extremely affected by drought and salt stress. Sweet potato is mainly planted in barren areas, therefore it is necessary to identify candidate genes for improving drought and salt tolerance [36]. To date, the functions of R2R3-MYB TFs from sweet potato have remained large unknown. In the current study, transgenic sweet potato overexpression *IbMYB1* improved salt tolerance, with enhanced expression of structural genes in the flavonoid biosynthesis [37]. Furthermore, overexpression *IbMYB116* in *A. thaliana* enhanced transgenic plants’ drought tolerance by up-regulating jasmonic acid biosynthesis genes and activating the ROS-scavenging system [38]. *IbMYB48* act as a positive regulator of drought and salt response in transgenic Arabidopsis [39]. Sweet potato is mainly planted in barren and remote areas, which are frequently exposed to drought and salt stress, so it is beneficial for improving drought and salt tolerance.

The MYB protein family has been widely reported to be involved in abiotic stress processes in plants. However, the specific role of sweet potato MYB330 protein in regulating drought and salt stress remains largely unknown. Therefore, it is very valuable to clarify the function of MYB330 protein under drought and salt stress. In this study, an novel R2R3-MYB gene, *IbMYB330,* was cloned from a drought-tolerant sweet potato cultivar Eshu11. The expression levels of *IbMYB330* were induced by drought and salt stresses. Overexpression of *IbMYB330* in tobacco significantly enhanced drought and salt tolerance. The levels of proline, superoxide dismutase (SOD), ascorbic acid peroxidase (APX), and peroxidase (POD) activities were significantly higher in transgenic plants under drought and salt stresses, while the content of malondialdehyde (MDA) was markedly lower. Overexpression of *IbMYB330* up-regulated genes involved in the reactive oxygen species (ROS) scavenging system, thereby improving drought and salt resistance in transgenic tobacco. This study provides information for further investigations into the function of *IbMYB330* in response to abiotic stresses in sweet potato.

## 2. Materials and Methods

### 2.1. Plant Materials and Growth Condition

The drought-tolerant sweet potato cultivar Eshu11 was employed for cloning *IbMYB330*. *Nicotiana tabacum* cultivar Wisconsin38 (W38) was used as wild-types (WT) and to characterize the function of *IbMYB330*. Sweet potato plants were cultivated in a greenhouse (28 °C, 16/8 h light/dark period). Transgenic and wild-types W38 tobacco plants were cultured on Murashige & Skoog (MS) medium in a greenhouse (22 °C, 16/8 h day/night cycle).

### 2.2. Cloning of IbMYB330 and Its Promoter

Total RNA was extracted from leaves of 90-day-old field-grown Eshu11 in accordance with the manufacture’s specifications of the plant total RNA extraction kit (Tiangen, Wuhan, China). Genomic DNA of Eshu11 was extracted by CTAB method [40]. The quality and quantity of RNA were visualized by 1% agarose gel electrophoresis, and quantification at optical densities of 260 and 280 nm by Nano-Drop ND-1000 spectrophotometer (Thermo Fisher Scientific, Wilmington, MA, USA). The cDNA samples were synthesized using *EasyScript* First-Strand cDNA Synthesis SuperMix (TransGen, Wuhan, China). Each 20 µL contained 4 μL 5 × *EasyScript*^®^ Uni All-in-One SuperMix, 1 µL gDNA remover, 1 μg RNA, and RNase-free water. The PCR process followed at 42 °C for 15 min, then 80 °C for 5 s. Specific primers *IbMYB330*-F/R (Appendix A) were designed by Primer Premier 5.0 to clone the DNA sequence of *IbMYB330*. The PCR procedure comprised an initial preheating step at 94 °C for 5 min, followed by 35 cycles of denaturation at 94 °C for 30 s, annealing at 55 °C for 30 s, and extension at 72 °C for 90 s, with a final extension at 72 °C for 10 min. PCR amplification products were separated by 1% agarose gel electrophoresis, and the target DNA fragments were recovered using *EasyPure*^®^ Quick Gel Extraction Kit (TransGen, Wuhan, China) according to the manufacturer’s instructions. The fragments were cloned into the pMD19-T vector (TakaRa, Wuhan, China) and sequenced by TIANYI Company (TIANYI, Wuhan, China).

*IbMYB330*-F/R primers were used to amplify the genomic DNA of *IbMYB330* with the following reaction procedure: initial preheating step at 94 °C for 5 min, followed by 35 cycles of denaturation at 94 °C for 30 s, annealing at 55 °C for 60 s, and extension at 72 °C for 90 s, with a final extension at 72 °C for 10 min. The PCR reaction system contained 5 µL 10 × PAGE buffer (Mg^2+^), 8 μL dNTPs, 1 μL *IbMYB330*-F/R primers, 0.5 μL *EasyTaq*^®^ polymerase, 2 μL genomic DNA, and 32.5 μL ddH_2_O. The PCR products were separated via electrophoresis on a 1% agarose gel, and the target DNA fragments were recovered using an *EasyPure*^®^ Quick Gel Extraction Kit (TransGen, Wuhan, China) according to the manufacturer’s instructions. The resulting fragments were cloned into the pMD19-T vector (TakaRa, Wuhan, China) and sequenced by TIANYI Company (TIANYI, Wuhan, China).

The sequence of *IbMYB330* promoter was cloned using the genomic walking method. The specific primers *IbMYB330*pro-F/R were designed according to the upstream sequence of *IbMYB330* (Appendix A). The PCR reaction system contained 5 μL 10 × PAGE buffer (Mg^2+^), 8 μL dNTPs, 1 μL primer *IbMYB330*pro-F/R, 0.5 μL *EasyTaq*^®^ polymerase, 2 μL genomic DNA, and 32.5 μL ddH_2_O. PCR amplification products were separated via electrophoresis on a 1% agarose gel, and the target DNA fragments were recovered using an *EasyPure*^®^ Quick Gel Extraction Kit (TransGen, Wuhan, China). The resulting fragments were cloned into pMD19-T vector (TaKaRa, Wuhan, China) and sequenced by TIANYI Company (TIANYI, Wuhan, China).

### 2.3. Bioinformatics Analysis

We conducted comparison and identification on the sequence of *IbMYB330* in the BLAST program of National Center for Biotechnology Information (NCBI) database (https://blast.ncbi.nlm.nih.gov/Blast.cgi?PROGRAM=blastn&PAGE_TYPE=BlastSearch&LINK_LOC=blasthome accessed on 25 September 2023). The IbMYB330 theoretical isoelectric point, molecular weight, and instability index were calculated with the online website resource ExPASy (https://web.expasy.org/protparam/ accessed on 25 September 2023). DNAMAN software (version 8.0, Lynnon Biosoft, USA) was employed to align the IbMYB330 and other plants. The multiple sequence alignments were performed by MEGA software (version X), and the phylogenetic tree was constructed followed by the neighbor-joining method. The online website MEME was used to analyze the protein motif (https://meme-suite.org/meme/tools/meme accessed on 26 September 2023). The distribution of *Cis*-acting elements in *IbMYB330* promoter was explored on PlantCARE (http://bioinformatics.psb.ugent.be/webtools/plantcare/html/ accessed on 25 September 2023).

### 2.4. Expression Analysis of IbMYB330

Research the expression profiles of *IbMYB330* in different tissues of sweet potato, stems, leaves, and roots. RNA was extracted from the sweet potato samples. The qRT-PCR were performed by Bio-RAD CFX96 (Bio-RAD, Foster City, CA, USA, version 1.1). Specific primers RT-*IbMYB330*-F/R (Appendix A) were used for qRT-PCR amplification. The expression profiles of *IbMYB330* in sweet potato under abiotic stresses were analyzed. Four-week field-grown Eshu11 were cultured in half-Hoagland solution with 200 mM NaCl, 20% polyethylene glycol-6000 (PEG-6000), and 20% H_2_O_2_, respectively. Sweet potato leaves were collected at 0, 1, 3, 6, 12, and 24 h after treatment and quickly frozen in liquid nitrogen according to Zhang et al. [41]. According to the manufacturer’s instructions, *TransStart*^®^ Green qPCR SuperMix (50 × 20 μL reactions) includes 2 × *TransStart*^®^ Green qPCR SuperMix, Passive Reference Dye (50×), and nuclease-free water. Each 10 μL mixture contained 5 μL *TransStart*^®^ Green qPCR SuperMix (TransGen, Wuhan, China), 0.4 μL of each specific primer, 3.2 μL nuclease-free water, and 1 μL cDNA. The qPCR program comprised preheating at 94 °C for 2 min, followed by 45 cycles of denaturation at 94 °C for 5 s, and annealing at 58 °C for 30 s. The expression levels of *IbMYB330* were detected using qRT-PCR analysis conducted on the 7500 Real-Time PCR system (Applied Biosystems, Foster City, CA, USA). There were three biological replicates for each experiment and three technical replicates for each sample. The genes’ relative expression levels were counted with the comparative C_t_ method [41]. *IbActin* and *NtActin* genes were employed as the internal reference genes of sweet potato and tobacco, respectively (Appendix A).

### 2.5. Construction of Overexpression Vector

*IbMYB330*-F1/R1 primers (Appendix A) were used to amplify the ORF of *IbMYB330*, and the coding region was inserted into the *Kpn*I and *BamH*I sites of plant expression vector pCAMBIA1300 to produce the *IbMYB330*-GFP fusion construct driven by the cauliflower mosaic virus (CaMV) 35S promoter (Appendix A). The resulting pCAMBIA1300-*IbMYB330*-GFP overexpression plasmid was confirmed by sequencing and then transformed into *Agrobacterium tumefaciens* (strain GV3101 competent cells) to further explore the function of *IbMYB330*.

### 2.6. Production of Transgenic Plants

Transgenic tobacco overexpression lines (OE lines) were obtained by transferring the pCAMBIA1300-*IbMYB330*-GFP recombinant plasmid into wild-type tobacco via the leaf disc method [41]. Infected tobacco leaf discs were inoculated on MS medium containing 15 mg/L Hyg, 400 mg/L cephalexin, 1.0 mg/L 6-BA, and 0.1 mg/L NAA in the dark at 27 ± 1 °C for 30 d, and then the regenerated shoots were transferred to 1/2 MS medium with 25 mg/L Hyg and 100 mg/L Carb for the formation of whole plants. Primers 35S-F/1300-R (Appendix A) were used to detect *IbMYB330* overexpression in tobacco plants. Genomic DNA was extracted from the leaves of transgenic and WT plants according to the CTAB method [40]. The PCR procedure comprised an initial preheating at 95 °C for 5 min, followed by 35 cycles of denaturation at 95 °C for 30 s, annealing at 62 °C for 30 s, extension at 72 °C for 1 min, and finally extension at 72 °C for 10 min. The PCR products were detected by 1% agarose gel electrophoresis to confirm the insertion of *IbMYB330* into the transgenic plants. Further, qRT-PCR was used to detect the expression of *IbMYB330* in transgenic plants, with the *NtActin* gene used as the internal reference gene of *Nicotiana tabacum* [42]. Transgenic lines were selected for phenotypic investigation and WT tobacco was used as the control.

The drought and salt tolerance of the transgenic plants were identified by the method of Zhang et al. [41]. The transgenic plants and WT plants were grown in MS medium with 200 mM NaCl to determine the salt tolerance of transgenic lines. To test the drought tolerance of the transgenic lines, WT and transgenic plants were grown in MS medium with 20% PEG-6000. The culture conditions were 26 °C, 16/8 h day/night cycle. After being cultured with treatment for four weeks, the growth status of the transgenic plants was observed, and the content of proline and protein and the activity of catalase (CAT), malondialdehyde (MDA), and peroxidase (POD) were determined. The expression levels of the abiotic stress-responsive genes were determined in the transgenic tobacco plants both under normal conditions and under treatment. The abiotic stress-responsive genes included *SOD*, *POD*, *APX*, and the proline synthesis-related gene, *P5CS*. The genes’ specific primers were designed by Primer Premier 5. The primer sequences are shown in Appendix A.

### 2.7. Statistical Analysis

IBM SPSS Statistics 26 software was used for data analysis, and Duncan’s multiple range tests were employed to analyze the significant differences. *p* < 0.05 and *p* < 0.01 represent significant and extremely significant, respectively. Experimental was set in 3 biological and 3 technical replicates, and data are presented as the means ± standard deviations.

## 3. Results

### 3.1. Isolation and Characterization of IbMYB330

In a previous study, a differentially expressed gene under drought stress treatment was found in the transcriptome database. According to the reference sequence, the full length of *IbMYB330*’s open reading frame (ORF) from Eshu 11 was cloned. The *IbMYB330* was 807 bp and encodes a protein with 268 amino acids. The molecular weight of IbMYB330 was 29.24 kD, and the theoretical isoelectric point and instability index were 9.12 and 67.91, respectively. Prediction of subcellular localization revealed that the IbMYB330 protein of sweet potato was predominantly distributed in the nucleus. The 1404 bp genomic sequence of *IbMYB330* consisted of three exons and two introns, which is similar to the exon-intron pattern of tobacco *NtMYB330* [30] (Figure 1a).

Multiple sequence alignment analysis showed that the IbMYB330 amino acid sequence was highly homologous to the MYB330 or MYB330-like protein of *Ipomoea triloba*, *Citrus sinensis, Ipomoea nil*, *Nicotiana attenuata,* and *Solanum tuberosum* (Figure 1b), showing that MYB330 is conserved in plants. IbMYB330 has the common characteristics of R2R3-MYB transcription factors. The N-terminal amino acids are highly conserved and contain two SANT domains (R2, R3), and the C-terminal amino acid sequence is diverse. A phylogenetic tree of IbMYB330 with *A*. *thaliana*, *Oryza sativa*, *Zey mays*, *I*. *triloba*, *I*. *nil*, *G*. *hirsutum*, *Gossypium raimondii*, *S*. *tuberosum*, and *Nicotiana tabacum* was constructed by MEGA X. It has been shown that MYB proteins are classified into six major categories; IbMYB330 belonged to the II subfamily and was closely related to StMYB1 (Figure 1c). The promoter sequence of IbMYB330 gene was analyzed (Figure 1d). The *IbMYB330* promoter region was analyzed. The *IbMYB330* promoter region contained multiple *cis*-acting elements associated with resistance, such as MYB, TCA, and TGA-element (Figure 1d, Appendix A).

The motifs of IbMYB330 and other plants in II subfamily were identified (Figure 2). Sequence analysis revealed motifs length ranging from 8 to 50 amino acids; motif 1 contained the complete structure Myb-like DNA-binding domain (R2), and motif 2 contained Myb-like DNA-binding domain (R3). Motifs 3, 4, 5, and 6 constitute a relatively complete conserved domain of R2R3-MYB transcription factors. The conserved motifs of II family members were analyzed. The results shown that motif 3 is duplicated in ZmMYB1 and motif 6 is absent in AtMYBB6; it is speculated that MYB members of different plants produce or lose specific conserved motifs in the process of evolution, and their functions are differentiated.

### 3.2. Expression Analysis of the IbMYB330 in Sweet Potato

The expression patterns of *IbMYB330* in sweet potato were analyzed by qRT-PCR, which showed that *IbMYB330* was expressed in the root, stem, and leaf tissues of sweet potato, and the expression varies among tissues. Expression levels of *IbMYB330* in the root were significantly higher than that in the leaf and stem (Figure 3).

### 3.3. Expression Profiles of IbMYB330 under Abiotic Stresses

To explore the effect of stresses on *IbMYB330* expression, qRT-PCR was used to determine the transcription level of *IbMYB330* in sweet potato under different abiotic stresses, namely drought (20% PEG-6000), salt (200 mM NaCl), and H_2_O_2_ (20%). The results showed that the expression of *IbMYB330* in sweet potato was strongly induced by PEG-6000, NaCl, and H_2_O_2_. Under 20% PEG-6000 treatment, the expression of *IbMYB330* reached a peak at 1 h, then its expression began to decline and remained higher than that at 0 h (Figure 4a). Under NaCl treatment, the expression of *IbMYB330* showed an up-regulated increase from 0 h to 6 h and reached the maximum at 6 h, and the expression of *IbMYB330* was higher at 1–24 h than at 0 h (Figure 4b). The expression of *IbMYB330* was induced by H_2_O_2_: the expression level reaches its maximum at 6 h induced, and it shows an up-regulation trend in expression (Figure 4c).

### 3.4. Overexpression of IbMYB330 Improves Tolerance to Drought and Salt Stress in Transgenic Plants

To gain insight into the function of the *IbMYB330*, the overexpression vector pCAMBIA1300-*IbMYB330* was constructed and transferred into wild-type *Nicotiana tabacum*. The leaf DNA from proposed transgenic and wild-type tobacco were used as templates, and the *IbMYB330* overexpression lines were identified by PCR with WT plants, *IbMYB330* plasmid, and ddH_2_O used as controls (Appendix A). There *IbMYB330* overexpression were selected for further research. The wild-type were used as controls. Three overexpression transgenic lines with high expression levels were selected for the next further study (OE-1, OE-8, and OE-13; Figure 5). The relative expression levels of OE-1, OE-8, and OE-13 were 5.344 ± 0.246, 7.945 ± 0.120, and 9.093 ± 0.165, respectively. The WT plants were used as control.

To determine the response of *IbMYB330* overexpression lines to drought and salt stresses, the three transgenic lines (OE-1, OE-8, OE-13) and the WT plants were grown in MS medium with 200 mM NaCl, 20% PEG-6000, and stress-free conditions for 4 weeks, respectively. There was no significant difference in growth was observed between the transgenic plants and WT plants under stress-free, while the transgenic lines had better growth status than WT plants (Figure 6a).

To further evaluate drought tolerance, transgenic lines and WT plants were cultured in MS medium with 20% PEG-6000. After 4 weeks, the growth status of transgenic lines was significantly better than that of WT plants (Figure 6a). Stress-related physiological indicators such as MDA content, CAT activity, POD activity, proline content, and soluble protein content, which reflected the defensive capability of plants, were measured in WT plants and transgenic lines under normal and stresses treatments. Under drought treatment, the MDA content in transgenic lines and WT plants was raised compared with that under normal treatment, and the content of MDA in the three transgenic (29.670 ± 0.605, 26.683 ± 1.958, 27.509 ± 0.565 nM g^−1^ · FW, respectively) was significantly lower than that in the WT plants (41.135 ± 1.809 nM g^−1^ · FW) (Figure 6d). Moreover, CAT activity in the three transgenic lines (3.670 ± 0.486, 3.387 ± 0.207, 4.657 ± 0.318, respectively) was significantly higher than that in WT plants (1.484 ± 0.104) under drought treatment (Figure 6b). Under drought treatment, POD activity was significantly enhanced in the three transgenic lines compared with that in the WT plants, and OE-1 and OE-13 had the similar POD activity (93.837 ± 1.367, 93.776 ± 0.811 10^3^ U g^−1^, respectively). Proline content in plants reflects the stress resistance of plants to some extent, and varieties with strong drought resistance tend to accumulate more proline [42]. Under the condition of drought, the proline in the three transgenic lines increased significantly. The proline contents in transgenic lines OE-1 and OE-13 were a similar level, and all were higher than that in WT plants (Figure 6e). Soluble protein content in crops is one of the important indicators of their resistance to stress [43]. The soluble protein content of transgenic lines changed significantly in response to drought stress, significantly higher than WT plants (Figure 6f).

Overexpression of *IbMYB330* in tobacco enhances the resistance of transgenic lines in response to salt stress. After being cultured in MS medium with 200 mM NaCl for 4 weeks, the growth of the transgenic lines was significantly better than WT plants (Figure 6a). Under salt stress treatment, CAT activity, SOD activity, POD activity, and proline content were significantly higher in the transgenic strain than that in the WT plants, and MDA content was significantly lower than in WT plants (Figure 6b–d). The content of proline and soluble protein changed under salt stress [44], and that in the transgenic lines was significantly higher than that in the WT plants (Figure 6e–f).

Under drought and salt stress treatment, the expression levels of the abiotic stress-responsive genes in transgenic lines significantly increased. Under drought and salt stress, ROS scavenging-related genes *SOD*, *POD* and *APX* were up-regulated in the transgenic lines compared with the WT plants, and proline synthesis-related gene *P5CS* was up-regulated (Figure 7a–d).

## 4. Discussion

Drought and salt stress are the most important abiotic stresses that threaten the normal growth of plants. In plants, regulatory protein transcripts of different families, including the AP2/EREBP, NAC, and the zinc finger protein family, are induced by abiotic stress. These transcription factors may regulate signal transduction pathways under drought, cold, or high-salinity stress, and may also regulate stress response gene expression to achieve stress tolerance [45,46]. The MYB TF family contains four subfamilies and is the largest class of plant TF, which plays a pivotal role in the control of a variety of plant physiological and biochemical processes. Among them, the R2R3-MYB transcription factors, an important role in response to various environmental stresses, have always been a research hotspot. At present, a large number of R2R3-MYB transcription factors have been reported to play vital roles in response to drought and salt stress [39,47,48,49]. To date, the role of *IbMYB330* in plants is still superficial.

In this study, we cloned the *IbMYB330* gene from the drought-tolerant sweet potato cultivar Eshu11. Its ORF encoded a polypeptide of 268 aa. The expression levels of *IbMYB330* was significantly up-regulated under PEG and NaCl stress treatment (Figure 4). Phylogenetic analysis revealed that IbMYB330 was classified in the group II and more closely related to *I. triloba* ItMYB308 (Figure 1c). Sequence multiplex analysis indicated that the IbMYB330 protein contains two Myb-like DNA binding domains (R2, R3), which revealed that it is a typical R2R3-MYB transcription factor (Figure 2). Its overexpression significantly improved the drought and salt tolerance of the transgenic tobacco plants (Figure 6). This study reveals, for the first time, the key role of *IbMYB330* in plant drought resistance and salt tolerance.

Studies have shown that gene expression in plants is tissue-specific [50]. In the present study, *IbMYB308*, *IbMYB116*, and *IbBAM1.1* had very similar expression patterns in sweet potato, and expression levels were significantly higher in leaves than in tissues such as roots and stems [38,51,52]. The relative expression levels of *IbMYB330* in sweet potato have tissue specificity: the expression level in the root is higher than that in the stem and leaf. This gene is mainly effective in the root and enhances plant stress resistance.

In response to abiotic stress, plants often produce a large amount of reactive oxygen species (ROS), including superoxide ions (O_2_^−^), hydrogen peroxide (H_2_O_2_), and hydroxyl radicals (HO^−^) [53,54,55]. ROS can act as an important signaling pathway molecule for plant defense, and its excess will lead to deleterious impacts on biological molecules including nucleic acids, proteins, lipids and other macromolecules which result in cellular damage and death [56]. ROS scavenging systems such as superoxide dismutase (SOD), ascorbic acid peroxidase (APX), peroxidase (POD), and catalases (CAT) can detoxify ROS to reduce oxidative damage in plant cells and enhance stress resistance [57,58]. Previous studies have shown that increased ROS-scavenging activity can contribute to enhance drought and salt tolerance [59,60,61,62]. It has been found that the accumulation of proline can provide the plant with stronger resistance to stress [63,64]. Many studies have shown that proline can regulate the pH of plant cytoplasm to prevent its acidification, protect membrane integrity, and also provide the function of clearing ROS [65,66,67,68]. The higher content of malondialdehyde (MDA) may cause certain damage to membrane and cells, which will lead to a series of physiological and biochemical reactions, and the drought and salt tolerance of plants will also be weakened [69,70]. Many MYB TFs were found to be involved in drought and salt tolerance in plants by regulating the expression of ROS-related genes. In *A. thaliana*, the overexpression of *LcMYB1* enhanced the expression levels of *P5CS1* and improved crops’ salinity tolerance [71]. It was found that ectopic expression of *GmMYB118* gene increased tolerance to drought and salt stress by reducing the contents of ROS and MDA [72]. ROS-mediated mechanisms of drought and salt tolerance were also found in sweet potato; ectopic expression of *IbMYB48* in Arabidopsis results in increased SOD activity and the stress resistance of transgenic plants was significantly enhanced [39]. The *ZmMYB3R*-overexpression Arabidopsis plants show increased activities of the antioxidant enzymes CAT, POD and SOD under salt and drought treatments compared to WT plants [73]. Proline is an osmotic pressure regulator that responds to cellular dehydration. Under drought, plants accumulate proline to defend themselves against adversity.

Thus, to evaluate and dissect the molecular mechanisms and physiological roles of *IbMYB330* underlying plant tolerance, the *IbMYB330* gene was ectopically expressed in tobacco. In this study, under drought or salt stress conditions, the content of proline was increased in transgenic plants (Figure 6e). The expression of proline synthase gene *P5CS* was significantly increased in transgenic plants compared with WT under drought and salt treatments, which enhanced the stress resistance of the transgenic plants. The CAT activity and MDA content were changed in transgenic plants. In this study, CAT activity and MDA content were significantly increased and decreased in *IbMYB330*-overexpression transgenic plants compared to those in WT under drought and salt stresses, respectively (Figure 6b,d). The POD activity and soluble protein content were significantly increased in *IbMYB330*-overexpression transgenic plants compared to WT under drought and salt stresses (Figure 6c,f). The expressions of genes encoding *SOD*, *POD*, and *APX* members of the ROS-scavenging system were all increased in transgenic plants compared with WT (Figure 7a–c). The above results indicate that *IbMYB330* activates the genes of the ROS-scavenging system under drought and salt stresses and enhances drought and salt tolerance in transgenic plants.

In conclusion, these results reveal that proline biosynthetic pathway and ROS-scavenging system are activated by overexpression of *IbMYB330*, thus improving abiotic stress tolerance of transgenic plants (Figure 8). The results of our study on the function of sweet potato *MYB* gene will provide a reference for cultivating plants’ resistance to abiotic stress, whereas *IbMYB330* is only heterologously in tobacco for functional validation. In the future, the potential function of *IbMYB330* in abiotic stress tolerance will further investigated by advanced technologies and methodologies.

## 5. Conclusions

R2R3-MYB is one of the most widely distributed transcription factor families in plants. In this study, an R2R3-MYB transcription factor gene, *IbMYB330* was isolated from Eshu11. Furthermore, the expression characteristics and functional aspects of *IbMYB330* were thoroughly investigated. Expression of *IbMYB330* was induced by drought and salt. Ectopic expression of *IbMYB330* in tobacco contributed to an increase in proline content and reactive oxygen species-scavenging system activity, suggesting that *IbMYB330* modulates tolerance to drought and salt stress in tobacco. The results of this study indicate that *IbMYB330* plays an important role in response to drought and salt stress, and *IbMYB330* may be a potential gene for improving the tolerance of plants to abiotic stresses.

## Figures and Tables

**Figure 1 genes-15-00693-f001:**
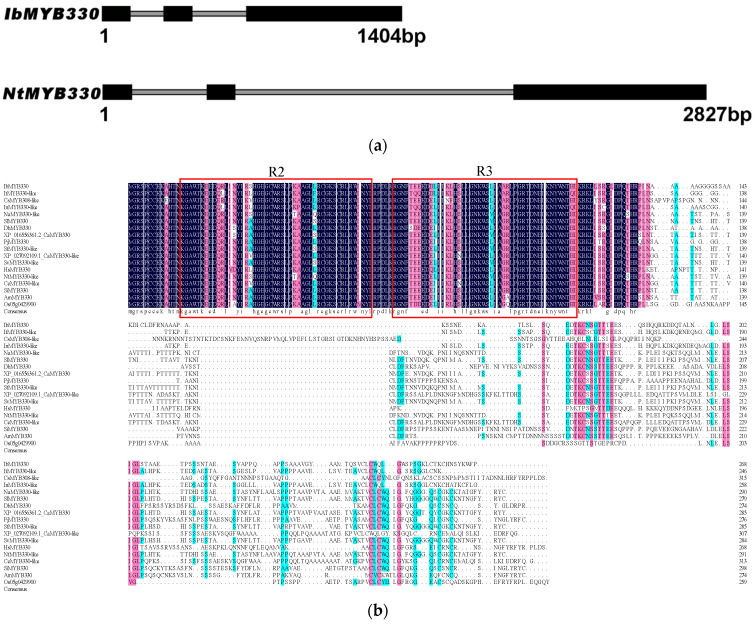
Sequence analysis of *IbMYB330*. (**a**) Genomic structures of *IbMYB330* and *NtMYB330*. Exons are represented by boxes and introns by lines. (**b**) Sequence alignment of *IbMYB330* with its homologues from other plant species. The R2 and R3 repeats represented by red box, the same amino acids represented by black box and different amino acids represented by blue box. (**c**) Phylogenetic analysis of *IbMYB330* and MYB transcription factors from *A. thaliana* (
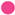
), *O. sativa* (
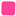
), *Z. mays* (
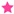
), *I. triloba* (
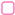
), *I. nil* (
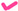
), *G. hirsutum* (
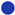
), *G. raimondii* (
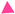
), *S. tuberosum* (
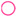
), and *N. tabacum* (
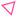
). (**d**) Cis-acting elements in the promoter of *IbMYB330*.

**Figure 2 genes-15-00693-f002:**
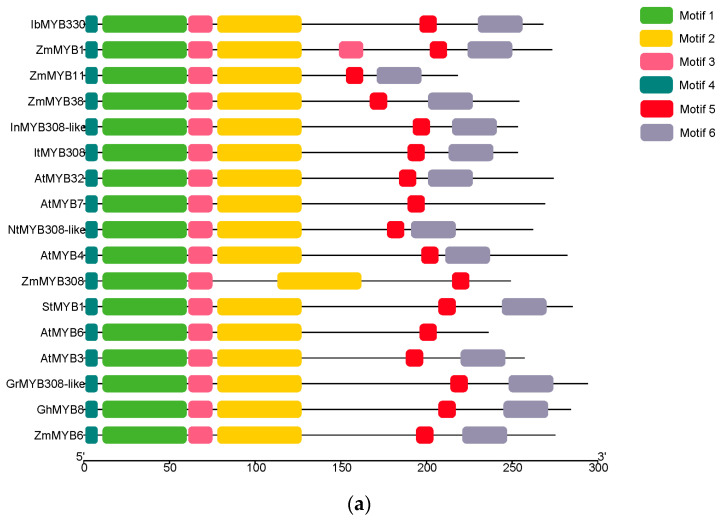
Motif distribution in *IbMYB330* and the II subfamily of the MYB family. (**a**) Motif locations. (**b**) The width and amino acid sequence of conservative motifs.

**Figure 3 genes-15-00693-f003:**
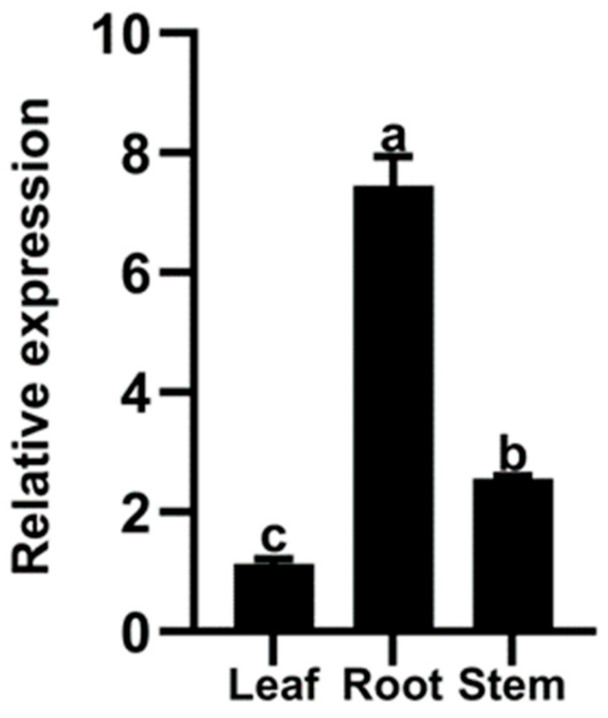
Expression of *IbMYB330* in different tissues of sweet potato by qRT-PCR. Data represent the mean of three biological replicates ± SD (*n* = 3). Error lines indicate standard deviations. Different lowercase letters (a–c) on the bars indicate significant differences at *p* < 0.01.

**Figure 4 genes-15-00693-f004:**
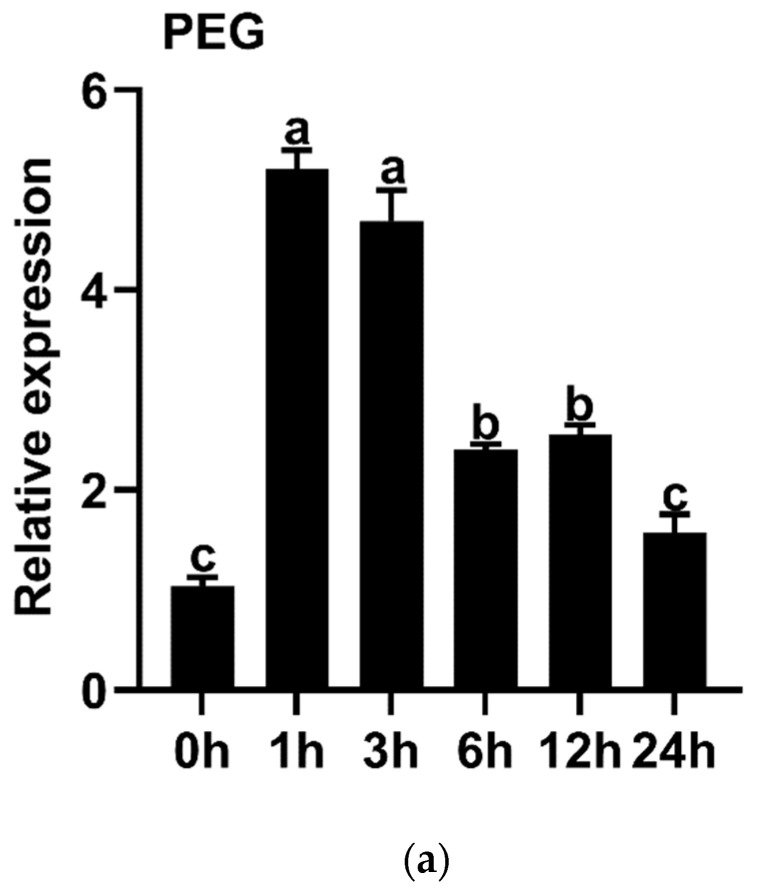
Expression profiles of *IbMYB330* in response to abiotic stress treatments. Sweet potato under (**a**) 20% PEG-6000, (**b**) 200 mM NaCl, and (**c**) 20% H_2_O_2_. *IbActin* was used as an internal reference gene. Data are presented as the means of three biological replicates ± SD (*n* = 3). Error lines indicate standard deviations. Different lowercase letters on the bars indicate significant differences at *p* < 0.01.

**Figure 5 genes-15-00693-f005:**
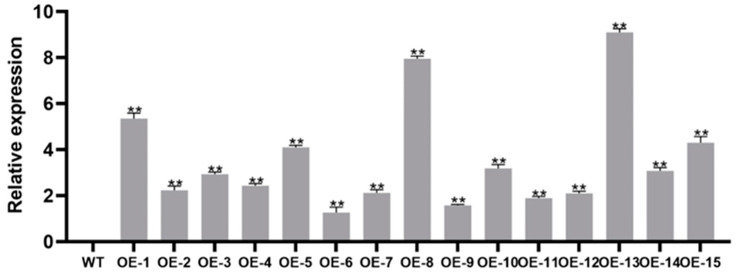
Expression analysis of *IbMYB330* gene in transgenic tobacco plants by qRT-PCR. Data are presented as the means of three biological replicates ± SD (*n* = 3). Error lines indicate standard deviations. ** on the bars indicate significant differences at *p* < 0.01.

**Figure 6 genes-15-00693-f006:**
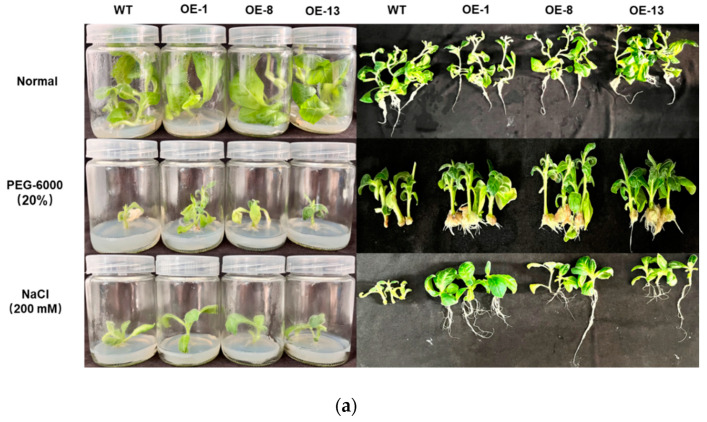
*IbMYB330* enhances drought and salt tolerance in transgenic plants. (**a**) Performance of *IbMYB330* transgenic plants and WT plants cultured for 4 weeks on MS medium without stress or with 20% PEG-6000 or 200 mM NaCl, (**b**) CAT activity, (**c**) POD activity, (**d**) MDA content, (**e**) proline content, and (**f**) soluble protein content. Data are presented as the means of three biological replicates ± SD (*n* = 3). Error lines indicate standard deviations. ** on the bars indicate significant differences at *p* < 0.01.

**Figure 7 genes-15-00693-f007:**
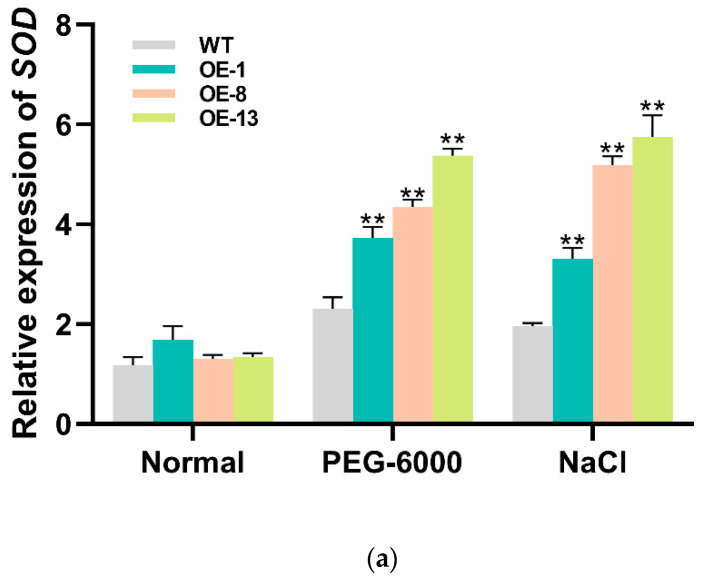
Relative expression of abiotic stress-responsive genes in the transgenic and WT plants. The abiotic stress-responsive genes of (**a**) *SOD*, (**b**) *POD*, and (**c**) *APX* and the proline synthesis related gene (**d**) *P5CS*. *NtActin* was used as the internal reference gene. Data are presented as the means of three biological replicates ± SD (n = 3). Error lines indicate standard deviations. ** on the bars indicate significant differences at *p* < 0.01.

**Figure 8 genes-15-00693-f008:**
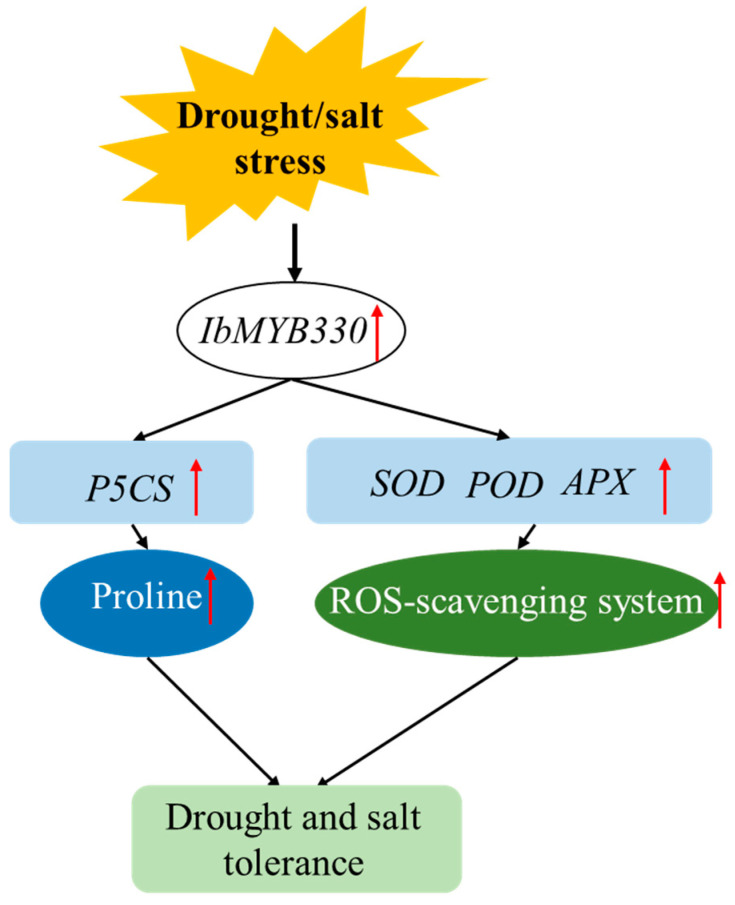
Proposed working model of the *IbMYB330* regulatory module in abiotic stress responses. The red arrows indicate elevated genes, proline content, and reactive oxygen scavenging system capacity.

## Data Availability

Data are contained within the article.

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
