# Peer review of "A Sweet Potato MYB Transcription Factor IbMYB330 Enhances Tolerance to Drought and Salt Stress in Transgenic Tobacco"

_genes, 2024, doi:10.3390/genes15060693_

Round 1
Reviewer 1 Report
Comments and Suggestions for Authors
1. English writing should be substantially improved.
2. Introduction, line 68. Sweet potato is not a cereal crop.
3. Introduction, lines 85-89. This paragraph should be rewritten, clearly indicating the objectives of the study and some related hypotheses. In the form it is presented it seems rather a summary of the results and the importance of the study.
4. Materials and Methods, Statistical analysis, lines 207-210. Clearly specify the type of statistical analysis (e. g. analysis of variance) and complementary information such as sources of variation, the number of true (not biological) replications, experimental unit, sample sizes. Clarify why the authors used Duncan’s test and at the same time standard error. If posible, include the analysis of variance output in the section of Results to confirm statistical significnces.
5. Conclusions. Conclusions should be based exclusively on the results of this study; please avoid reporting here information extracted from the literature (e.g. line 428) or information corresponding to the Materials and Methods section (e.g. lines 429-431).
Comments on the Quality of English LanguageEnglish needs to be improved substantially. There are multiple typos and phrases with bad syntax.
Reviewer 2 Report
Comments and Suggestions for Authors
The authors investigate the function of one MYB gene from sweet potato by transferring it to Nicotiana tabacum. However, they failed to explain why they specifically focused on IbMYB330. Thus, Introduction should be rewritten by providing more background information.
Please find my other comments in the attached file.

There are lots of typos and grammatical errors. I strongly recommend this manuscript should be revised by a fluent English user who has a scientific background.
Round 2
Reviewer 2 Report
Comments and Suggestions for Authors
Introduction should be revised by clearly stating why IbMYB330 was focused in this study.
Comments on the Quality of English LanguageThere are still many grammatical errors.
